# What is the lived experience of anxiety for people with Parkinson's? A phenomenological study

Christopher J. Lovegrove[1,2]* , Katrina Bannigan[3]

**1** Royal Devon & Exeter NHS Foundation Trust, Exeter, United Kingdom, **2** School of Health Professions, Faculty of Health & Human Sciences, University of Plymouth, Plymouth, United Kingdom, **3** Department of Occupational Therapy and Human Nutrition and Dietetics, School of Health and Life Sciences, Glasgow Caledonian University, Glasgow, Scotland, United Kingdom

☯ These authors contributed equally to this work.
* christopher.lovegrove@nhs.net

## Abstract

### Purpose

Anxiety is a common non-motor symptom of Parkinson's and there is no specific pharmacological intervention for people with Parkinson's who experience anxiety. Yet there is little specific research documenting how individuals with this condition experience anxiety. It is important to explore the experiences of people with Parkinson's to identify potential issues in developing future non-pharmacological interventions. This study explored the lived experience of anxiety for people with Parkinson's.

### Materials and methods

Six participants were recruited into a descriptive phenomenological study, through the charity Parkinson's UK, using a maximum variation sampling strategy. Face to face interviews were conducted. Data analysis employed thematic analysis.

### Results

Three key themes encapsulated the data: Finding ways to cope to "Try not to let it rule your life", Amplifies symptoms "It's emotionally draining it it's also physically draining" and "Anxiety is a funny thing" experienced in myriad ways. A model of the experience of PWP experience of anxiety is proposed.

### Conclusions

Anxiety is a complex experience constructed of interlinked parts affecting people with Parkinson's in myriad ways. Researchers and healthcare professionals should take these findings into account when designing future studies and interventions.

**Data Availability Statement:** All relevant data are within the manuscript and its Supporting information files.

**Funding:** CL- This research was supported by the National Institute of Health Research (grant

number MRES-2015-03-013-105). https://www.
nihr.ac.uk/ The funders had no role in study design,
data collection and analysis, decision to publish, or
preparation of the manuscript.

**Competing interests:** The authors have declared
that no competing interests exist.

## Introduction

Parkinson's disease, commonly referred to as Parkinson's, is the second most common neuro-degenerative condition worldwide [1]. For example, in the UK, it affects approximately 145,000 people [2]. The majority of people with Parkinson's (PWPs) (98.6%) experience non-motor symptoms and, of these 43–56% experience stress and anxiety [3, 4]. Walsh and Bennett [5], in a comprehensive literature review, found a higher prevalence of anxiety in PWPs compared to the general population. These studies suggest that anxiety is more common in PWPs than previously believed [6]. This is of concern because PWPs who experience anxiety are more at risk of falling and experience a lower quality of life, loss of independence, loss of social roles, and increased health burden [7, 8]. It has been identified that excessive anxiety in PWPs results in a higher frequency of motor fluctuations [9, 10].

### Anxiety and Parkinson's

Hallion and Ruscio [11] describe anxiety as a persistent internal feeling of fear and worry that is intrusive in daily life. Walsh and Bennett [5] highlight that PWPs experience more anxiety than people with multiple sclerosis and, in particular, are afraid of being negatively evaluated in public. This suggests that these individuals experience anxiety differently to other populations, and that an earlier age of anxiety onset in Parkinson's could be an indicator of higher levels of anxiety.

Psychological stressors that are associated with long-term conditions and experienced by PWPs can increase anxiety [12]. However, PWPs may be more susceptible to anxiety than people with other long-term conditions due to Parkinson's-associated dopamine deficiency [13]. Dopamine is a modulator in the amygdala, a brain structure involved in fear and anxiety [14]. When dopamine is deficient, this produces neuronal hyper-excitability and exaggerated responses to perceived threats [14, 15]. Although PWPs can be treated with dopamine-replacement medication they often experience a marked increase in symptoms as the medication wears off throughout the day [16]; primary (or type-1) worry can in turn rapidly progress to type-2 worry (meta-worry, or 'worry about worry') [17]. This can further increase anxiety symptoms which contribute to the maintenance of the hyper-excited neuronal anxiety circuit [18].

The experience of anxiety in Parkinson's has also been thought to be related to age [19]. Whilst Barone et al. [3] found that approximately 40% of PWPs experienced anxiety that affected their everyday life they were unable to establish a link between the experience of anxiety and the age of the person with Parkinson's. Burn et al. [20] suggested that there is a strong relationship between the age of Parkinson's onset and anxiety, specifically that younger age of onset resulted in higher anxiety levels. Evidence suggests that younger PWPs are the primary population affected by anxiety [3, 5, 21]. Ehgoetz Martens et al. [21] found that an increase in anxiety resulted in an increase in freezing of gait (a brief, episodic loss, or reduction, of forward progression of the feet despite the intention to walk) episodes in both 'freezers' and 'non-freezers' ($p = <0.001$–0.013). These results suggest that anxiety is an important mechanism underpinning freezing of gait. The implication is, if PWPs are more anxious, they are likely to find mobilising more difficult and reduced mobility has a negative correlation with quality of life [22, 23]. This is a significant implication for clinicians and researchers to consider when developing interventions [24, 25], especially as there is no specific pharmacological intervention for PWPs who experience anxiety [26]. The link to motor performance was also noted in that the participants were highly anxious about falling, which has been supported by other literature [25, 27]. Other potential sources of anxiety for all older people, not just those with Parkinson's,

include health, families, work, finances, lightning, and heights; all of which can change over the life course [28].

Research suggests that anxiety is not experienced as a singular entity but as a set of complex multi-dimensional interactions with the world [29, 30]. Whilst these studies provide insights, generally there is a lack of anxiety-specific research in Parkinson's [3, 10, 31, 32]. The relationship between anxiety and motor performance requires further research [9, 10, 21]. Anxiety has been identified as a research priority by the charity Parkinson's UK [31]. Given its prevalence, the lack of a specific pharmacological intervention, and the dearth of specific research, it is vital to gain an understanding of the experience of anxiety for PWPs to inform the development of future complex interventions [33]. A patient and public involvement consultation was undertaken to verify the need for this research with PWPs and to seek their opinions on how the study should be conducted [34]. All participants stated that they felt anxiety in Parkinson's needed more research, thus strengthening the rationale for this study. As no research exploring the lived experience of anxiety for PWPs was identified, and it has been identified as an important area for research, the aim of this study was to gain an in-depth understanding of the lived experience of anxiety for PWPs.

## Materials and method

The approach that guided this study was descriptive phenomenology—rooted in the philosophy of Edmund Husserl (1859–1938)—which recognises that peoples' descriptions of their real-world experiences are important, because they shape actions, and so merit being the subject of scientific investigation [35]. Essentially, phenomenology's purpose is to illuminate the essence of a person's experience in relation to a specific phenomenon; in this instance first-hand accounts of the experience of living with anxiety for people with Parkinson's [36, 37]. To this end, phenomenology typically uses small sample sizes because, if an understanding of individual lived experience is the objective, data saturation is irrelevant [38]. Equally, whilst quantifiable and generalizable conclusions are not the objective, there is potential for an exploration into individual experiences to offer insights into, and understanding of, the human condition [39]. A feature of phenomenological research is 'bracketing'; the putting aside of one's own beliefs and knowledge about the phenomenon to avoid personal biases and prejudices influencing data collection and analysis. Throughout this research process, the researcher used strategies outlined by Chan, Fung, and Chien to achieve this, such as mental preparation, i.e deciding the scope of the litertaure review, planning the data collection, and planning the data analysis [40]. Collectively using these steps throughout the study mitigated any potentially deleterious effects of preconceptions that may have occurred.

Participants in the patient and public consultation (described above) also provided practical recommendations for how the study should be conducted. These recommendations were used to directly improve the design of this research study [34]. Ethical approval for this study was granted through the University of Plymouth's Faculty of Health and Human Sciences (reference number (16/17)-244). All participants provided written consent. People with Parkinson's can experience cognitive deficits [41]. This has the potential to hinder their ability to give informed consent regarding individual decisions [42]. Thus, the researcher—an experienced occupational therapist familiar with working with people with cognitive impairments—assessed whether the individual participants had the capacity to make a decision to participate in the study. This was completed during the introductory phone call, based on the UK Mental Capacity Act, with a standardised joint-service mental capacity assessment document if required [42].

## Participants

Having secured ethical approval for the study, six participants were recruited using Parkinson's UK 'Research Support Network' [43]. A maximum variation sampling strategy was successfully used to recruit people with experience of Parkinson's across different stages of the condition. All of the participants felt they experienced anxiety that was either caused by, or related to, their Parkinson's. Three men and three women were recruited across the three main stages of the condition according to the Thomas and MacMahon staging (diagnosis/ early, maintenance, and complex) [44]. A man and a woman were recruited for each stage. The mean age of the sample was 68 years (range = 59–86). All participants were white. Five participants were married to, or in de facto relationships (those living together on a genuine domestic basis and are not legally married or related by family), all with partners of a different sex and gender, and one participant was widowed. Five participants were retired, and one participant maintained a part-time self-employed job. All participants were white-British people, as classified by the UK census [45] from the South West of England, UK. All of the participants were living in their own homes.

## Data collection

This research was conducted in accordance with American Psychological Associations' Journal Article Reporting Standards for Qualitative Research [46]. Interviews were held in the participants' homes in an area where they felt comfortable. The aim of this was to increase convenience and privacy as it is suggested, that if participants are comfortable, they are more likely to reveal the nature of their lived experience [47, 48]. During the interviews, the researcher referred to the interview schedule (S1 File). This was developed for the study by adapting the wording of the interviewing framework proposed by Bevan for phenomenolgical studies [49]. Questions were adapted to focus the participant thoughts towards the study phenomenon; for example "Please can you tell me about your diagnosis with Parkinson's" and "What is your experience of anxiety?". A semi-structured approach was utilised to allow participants to elucidate and explore matters considered important to them [47]. Using an interview schedule allowed the researcher to investigate the research question in each interview while giving participants the opportunity to illustrate their own 'lifeworld' [49–52]. A digital Dictaphone with encryption capability was used for all audio recording and field notes were taken during and immediately after each interview. The purpose of this was to aid in developing understanding and meaning of the studied phenomenon by documenting observed behaviours, thoughts, and feelings [53].

## Data analysis

Thematic analysis was used to develop a deep insight through systematic reflection that could help the emergence of meaningful order from the collected complex, rich data [54, 55]. This method provides a flexible but robust process of analysis using coding to identify ideas from raw data and then using these codes to identify patterns across a dataset [56]. Thus, a ´´´theme" can be described as the subjective meaning and cultural-contextual message of data. Codes with common reference points by which ideas can be unified throughout the study phenomenon can be transformed into a theme [57]. Following each interview, the researcher prepared the data for analysis by preparing a verbatim transcription of the encrypted audio recordings into a word-processed document, formatted to support the coding process. Each transcript was completed within 24 hours. Braun and Clarke [56, 58] emphasise the importance of transcription as the beginning of the data analysis process. The researcher reread the transcripts and listened to the audio recordings simultaneously, on several occasions. This was

part of the immersion phase, a process whereby the researchers immerse themselves in the collected data by examining the data in depth [59], to involve the researcher in the participant's experiences and increase their familiarity with the data. Following this, the researcher entered an incubation phase where they purposely stepped away from the research for one month to support bracketing and the emergence of tacit knowledge [40, 60]. The data analysis was managed using software NVivo v10 [61]; a qualitative data analysis software package that helps researchers to organize and analyze qualitative research data. NVivo is atheoretical and the researcher uses it in line with their philosophical position. The codes chosen used the participant's own words as closely as possible with the aim of supporting participant's voices to come through in the analysis [58, 62]. Following this, the codes were grouped into sub-categories and then themes.

## Trustworthiness

Trustworthiness refers to the degree of confidence in data, interpretation, and the methodology used to confirm the quality of a study [63, 64]. A number of strategies were used to ensure the analysis accurately reflected the participants' lived experiences. Firstly, during the generation of themes, the researcher underwent a peer review process with a senior researcher (KB) who reviewed and challenged the analysis to assure the rigor and credibility of the data analysis [53, 65, 66]. This constructive feedback and interrogation resulted in several themes being further collapsed or renamed. Next, all six participants were given an opportunity to view a summary of the findings to check for accuracy in a member checking process [63, 64]. The purpose of this was to avoid misrepresentation of the participant's views and ensure the data truly reflected their experiences rather that the researcher's beliefs and assumptions [67]. Four of the six participants responded; they agreed with the overarching themes and offered additional insights into each theme for the researcher to consider. In the creative synthesis phase themes were illustrated with vivid quotations from the interview transcripts to explore the experience of the phenomenon of anxiety in Parkinson's [53].

## Results and discussion

Three overarching themes encapsulated the lived experience of anxiety for PWPs

- Finding ways to cope to "Try not to let it rule your life"

- Amplifies symptoms "It's emotionally draining it it's also physically draining."

- "Anxiety is a funny thing" experienced in myriad ways.

The themes are presented in turn and are discussed in relation to the literature. Areas for research have been highlighted throughout. These findings are then integrated with the knowledge from the literature into a proposed model of the experience of PWPs experience of anxiety (Fig 1).

## Finding ways to cope to "Try not to let it rule your life"

'Coping'—a complex concept—has been defined as habitual and enduring patterns of behaviours displayed by a person when confronted by a situation needing a response [68]. All of the participants had had to find coping strategies to manage their anxiety, but they had different experiences of this. For example, participants described the benefits of sharing their experiences with other PWPs as positively influencing their wellbeing:

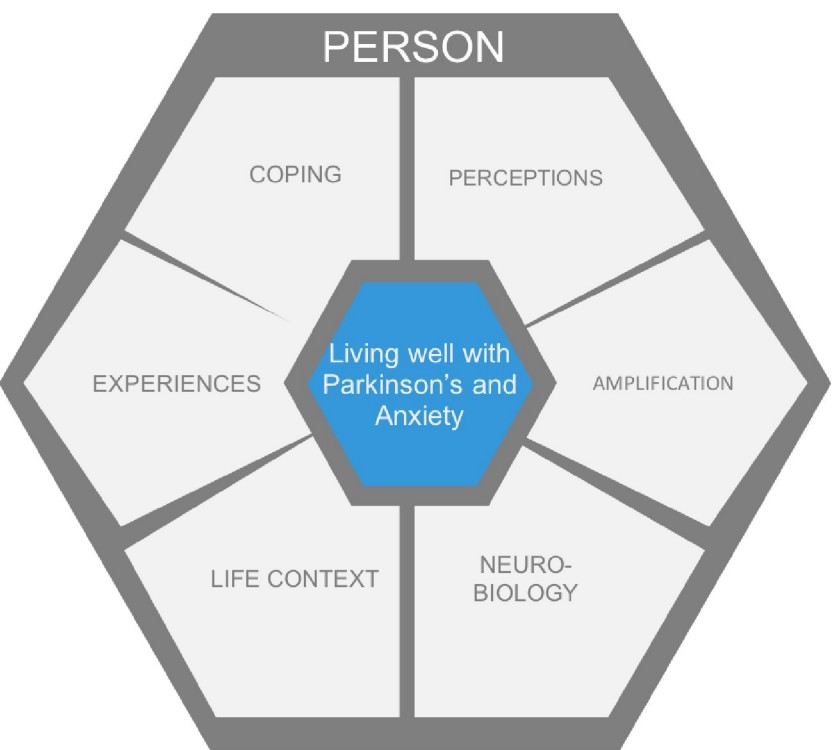

**Fig 1. Proposed model of the experience of PWP experience of anxiety.**

*"Which is very beneficial, a because it's six guys we started out with and we are close knit family if you want to call it that and that's good camaraderie and that's good for your own wellbeing and physical exercise has been proven in Parkinson's to to, to help and suppress the anxiety things"*

*(Alan, lines 168–174)*

*"Parkinson's is a big neurological disease. No one knows how it's going to pan out and there's a certain amount of nervousness about it. But if you're with other people who are similarly affected you can discuss, talk to themselves, their carers, their wives, their husbands. There's a few people younger than me not many. And we share experience. To be fair you become very friendly with them."*

*(Daniel, lines 282–290)*

Other participants shared negative feelings towards shared experiences of anxiety with other PWPs; some participants even valued experiencing their anxiety alone:

*"Soothing words, pfft, its that's not gonna help. So it doesn't help the person in front of me if there's someone there, and it doesn't really help me. Some say it clears out the stables it releases feelings, well for me something like that doesn't. It just makes it worse."*

*(Edward, lines 501–507)*

The participants' alternative coping strategies to manage their anxiety included being reliant on one's own abilities rather than those of others. Five participants highlighted the importance of self-reliance and self-confidence to them:

*"In the end I've got to be answerable for myself. Other people cannot be responsible for me. And I think you've got to have trust in yourself. Faith in yourself. In the long run it comes to you."*

*(Beth, lines 576–581)*

*". . . I just thought you've got to get on with it as much as you can, as well as you can. And try not to let it rule your life."*

*(Clare, lines 208–210)*

Some participants emphasised how they felt that the support of other people remained important in their anxiety experience:

*"I'm very lucky with my partner <name>. Whose granddad died, or had Parkinson's when he died. Erm so she's experienced it and will jokingly tell me don't be so stupid and buck up you know. She kicks me up the bum now and then if I'm feeling anxious or worried about something I shouldn't be worried about."*

*(Daniel, lines 257–267)*

As well as identifying active ways of coping with anxiety in Parkinson's, participants acknowledged that 'losing the ability to cope' was an important component of their lived experience of anxiety. Losing the ability to cope, and reduced locus of control, is associated with anxiety [69, 70], and the participants reflected this:

*"I feel a little bit like I have lost that capacity. And maybe perhaps, thinking about it, perhaps I am getting cross with myself that I can't deal with it as well as I used to."*

*(Clare, lines 625–629)*

## Amplifies symptoms "It's emotionally draining it it's also physically draining"

The second theme captures how anxiety exacerbated or amplified the participants pre-existing symptoms, including their engagement in public and social life, ability to think, their emotions, and self-identity, as well as their physical symptoms. This had an impact on their daily occupations and participation:

*"And-and you're trying inadvertently not to be conspicuous because it's not normal is it. It's not normal to be different at the meal table we'll say. Or it's not normal to be when you go to pick up a glass of orange squash or so or beer and you and, yeah."*

*(Alan, lines 673–679)*

*"So I now make sure, the way the way I engage with the outside world is predominantly in social context now. If I give a talk to my Probus group or my PD group, it's low profile it's not a big issue. It's something I've done before, it's something I can rehearse."*

*(Edward, lines 288–297)*

These quotations suggest that anxiety affected the participants' engagement in public and social life.

**Ability to think.** Participants openly expressed the experience of anxiety on cognition, and this was experienced in different ways in a variety of circumstances:

> "Yes I have always thought I was quite a clear thinker but if I'm pushed, or if I'm stressed now, I just, I just forget basic ordinary things. And I just can't, I just can't think logically I need to wait until I settle. And go back to a problem."

> *(Clare, lines 402–407)*

Similarly, Edward shared his experience of how anxiety affects his cognition and mimicked the physical symptom of freezing of gait:

> "And the thought process then gets, in the same way as you physically stuck I can get mentally stuck. I can only like it it's not a vicious circle because that indicates motion. The brain stops!"

> *(Edward, lines 420–425)*

**Emotions.** Participants described anxiety as having a negative effect on their emotions. The emotional anxiety experience was never described as positive; it amplified feelings of fear, nervousness, and derealisation:

> "My experience of it is, it can leave you nervous, it can make you nervous. It can make you very concerned about issues that might not be so difficult to deal with but they become a problem more than they really are."

> *(Alan, lines 526–530)*

> "Well I sort of, sweaty. And I feel, I feel I go starey. A bit like a rabbit in the headlights. Sometimes. And I just feel that I'm there but I'm not part of what's going on. I feel like I'm an outsider looking in, an observer of a situation rather than being part of it. . ."

> *(Clare, lines 459–465)*

**Self-identity.** During the interviews, the participants described experiences that suggest Parkinson's changes one's self-identity and relationship with other people. This represents an amplification of the social consequences of anxiety in Parkinson's. Often this experience was negative:

> "I was surprised when I had stopped working, that I missed the sense of purpose. I still do miss having something to get up for. Although I've got bits and pieces that I do, plenty to keep me busy but I still miss having that, goal because I' worked while 25 years. I was never previously off sick I was the sort of person who was never ill and worked most of the year without any days off sick. So it was a bit of a shock."

> *(Clare, lines 302–311)*

> "I just feel that I'm a different person and I'm afraid to go anywhere because I'm afraid I'll feel ill or something you know. . ."

> *(Gillian, lines 162–165)*

**Physical symptoms.**   Participants described how anxiety amplified their physical symptoms:

*"It manifested itself as these pains in my legs and they're not shaking so much I don't shake a lot I am at the moment this is because I'm talking to you. But that's not, its stiffness really. And my foot is turned out completely."*

*(Beth, lines 144–149)*

*"I think it makes me slower, speech wise it-it appears to change my gait when I get tired and anxious. My gait changes, my walking pattern changes. To small steps. That's probably it if I get anxious, if I get tired that happens as well. So, so they seem to be linked somehow. . ."*

*(Daniel, lines 241–247)*

### "Anxiety is a funny thing" experienced in myriad ways

The final theme captures the essence of the participants' experience of anxiety which they described in myriad ways. Most participants experienced anxiety as a negative, ubiquitous presence:

*"It's emotionally draining it it's also physically draining depending by the end of the week what sort of week you've had, you can just feel whacked by Friday lunchtime."*

*(Alan, lines 355 to 359)*

*"A sort of compression really I think. You feel everyone is watching you, you're looking around. Lack of confidence I think, perhaps comes in with anxiety as well. . ."*

*(Daniel, lines 330–333)*

*"I've lost confidence to do things you know. I hate it, I hate the way I am. I absolutely hate it."*

*(Gillian, lines 40–43)*

The causes of anxiety in Parkinson's were described in many different ways, relating to the participants' experience of physical symptoms and their impact on control and unfamiliarity:

*"There's anxiety there to a certain extent, am I going to do this properly? And sometimes the anxiety is really there when you're out-of-control and you've got no control of your body."*

*(Beth, lines 495–499)*

*"I got anxious about finding the seat. It was really strange statement I've not really had before. It was a strange stadium I didn't know where I was going. I was anxious of the loss, I was by myself and I was anxious about where my seat was."*

*(Daniel, lines 198–204)*

Some participants touched upon depression as part of their anxiety experience. Anxiety and depression are linked in the research literature and often coexist [19, 71, 72]. Participants described episodes of depression after diagnosis that passed, but did not identify themselves as currently feeling or being depressed:

*"Aaand yeah I've had times where I've felt depressed. And I haven't taken medication I mean I've always been one for, I've always been a one to make sure I don't look down that avenue I don't suffer with depression or haven't had depression."*

*(Alan, lines 158–163)*

The participants' experienced anxiety in Parkinson's as a ubiquitous, detrimental presence in their everyday lives:

*"And it's it's a cycle like that more or less everyday."*

*(Alan, lines 359–360)*

*"And that's an anxiety which is, it is always there but I would say up to now it's background. Background noise. I can see depending on the way the condition goes I could be more and more anxious about me and those around me."*

*(Edward, lines 323–328)*

## Discussion

The notion of sharing experiences between peers living through a common phenomenon, described by the participants, is an established support mechanism, i.e. peer support [73, 74]. Its usefulness may be explained by the psychology of upwards and downwards social comparison [75, 76]. Wills [77] explains downwards social comparison as a typically defensive propensity whereby the person looks at another individual that they consider worse-off to feel better about their situation.

*"I'm fine one-to-one and I'd much rather talk to people individually. Um, I'm not very happy within the group, I mean it's not a huge group there's usually only five or six of us but it does get a bit monotonous. . . Which sounds very negative and horrible, but there's a lot of repetition and people are very forgetful which is again something else that is probably typical."*

*(Clare, lines 558–568)*

Conversely, Collins [78] describes upwards social comparison as a person consciously or subconsciously comparing themselves to those that they perceive as better off or coping better, to create a more positive perception of their reality. Alan explained his experience of attending a Parkinson's exercise group, and how this affected his ability to cope:

*"Everybody can see what each other is doing and how succeed and fail in certain things, and you live off you feed off of each other and give each other confidence. And that's a big help. To know that there are other people with the same problem, getting by."*

*(Alan, lines 844–850)*

It may be that the participants in this study employed a combination of social comparison to manage their anxiety. As social comparison has been linked to positive and negative outcomes, including those related to anxiety, this may be an area of future research interest when developing a targeted Parkinson's anxiety intervention [75, 79].

Coping strategies for anxiety in PWPs are not addressed in the literature but the development of alternative coping strategies is not unique to the Parkinson's anxiety experience. The

different ways that participants had found to cope have been identified by other people, across cultures, living with a multitude of health conditions including anxiety [80, 81]. Finding ways to cope could be considered an expression of human response to crisis [82–84].

Self-reliance and self-confidence have been identified as essential components in managing anxiety; they empower people to seek out advice as well as implementing coping strategies [85]. However, it has also been noted that people experiencing anxiety demonstrate impaired cognitive information processing [86]; a known non-motor symptom of Parkinson's [87]. In conjunction with the other symptoms of Parkinson's, this can result in eroded self-confidence leading to people trying to reduce uncertainty in their lives and becoming more reliant on others [88]. This was reflected in the participants' experiences.

Participants reported that the support of other people was a necessary driver for overcoming anxiety. Relational support has been found to be important in stressful situations [29], having a protective role and assisting to reduce distress [89]. Conversely, this support may have potentially adverse consequences for PWPs managing anxiety, such as increased social avoidance due to reliance on others [89, 90]. Current anxiety–related research focused on resilience [91] could explain the importance of social support in managing anxiety. The variety of experiences voiced by participants means that further studies in this area could be beneficial.

Issues around coping and losing the ability to cope have previously been identified in PWPs [29, 30]. It could be argued that the younger participants were more likely to experience this as younger PWPs are more liable to experience severe anxiety [20]. However, this viewpoint is countered by Gillian. She is an older person, who feels she knows what she should be doing to feel better but is unable to cope with her anxiety.

*"And I know all the answers and I know that I should do more. Because the less you do the less you want to do. I've lost confidence to do things you know. I hate it, I hate the way I am. I absolutely hate it."*

*(Gillian, lines 38–43)*

Gillian's experience echoes those of other older adults [92, 93]. While a younger age of Parkinson's onset is likely related to an early commencement of anxiety symptoms [20, 94]. PWPs of all ages experience a loss of coping and control. This is a consideration for the development of future interventions [95]. The range of experiences of PWPs also point toward the need for a person-centred approach when developing an intervention.

Research has described the cognitive consequences of anxiety, particularly in spatial and working memory [96]. In Parkinson's, the relationship between anxiety and working memory has been noted particularly in PWPs with left hemibody onset [97]. It might be that participants are encountering an overload of their working memory contributing to their anxiety experience [98]. It is interesting that Edward described an experience that mirrored freezing of gait. It has been recognised that PWPs who experience freezing of gait episodes have dysfunction in brain areas that interact with mobility [99]. Edward describes the reverse of this experience, i.e. his brain freezes not his mobility, possibly identifying a novel avenue for future research.

Issues with emotional regulation involving dopaminergic pathways have been identified in PWPs [100], so this may be a contributing factor to the participants' experiences. This suggests that these occurrences are not isolated to PWPs. It may be argued that the underlying pathology of Parkinson's, coupled with the experience of living with an incurable neurodegenerative condition that affects participation in daily occupation, could amplify this experience [30, 101, 102].

Considering the strong link between anxiety and depression in PWPs in the quantitative literature [3, 32], it is interesting that participants' stories did not reflect this. It could be that the participants felt it was not relevant to the study, or the interview schedule required adapting to take this into account [48, 103]. It should be considered that anxiety could exist as a phenomenon unique to depression yet existing in a complex dyadic relationship [12, 101].

The participants expressed a sense of loss of control that has been identified in other experiences of living with a long-term condition [104, 105]. The loss of control equates to a sense of powerlessness, where individuals perceive that they have no influence over their life [106, 107]. Control becomes invested in external forces and fate [107]. Closely tied to powerlessness is 'meaninglessness'; an invasive feeling of the absence of significance, direction, and/or purpose [108]. Through a phenomenological lens, it could be seen that the lifeworld experience of anxiety in Parkinson's is one of increasing powerlessness and subsequent meaninglessness. With this in mind, future interventions to help PWPs live well with anxiety should focus on restoring *meaningfulness*.

Clare's experience of "*I just feel that I'm there but I'm not part of what's going on*" has been described in other studies exploring anxiety and derealisation in other populations [109, 110]. Derealisation is an altered perception or experience of oneself and their surroundings, resulting in the person feeling detached from the world that they participate in [111]. This could be interpreted as a protective experience; a disconnect from the anxiety stimulating constant worry [110]. While the findings of Sunvisson [29] touch upon some themes linked to Clare's experience of derealisation, this has not been explicitly investigated in the wider Parkinson's literature. This is relevant as it identifies another area of potential future research.

Self-identity is challenged on a daily basis through a variety of dynamic processes, such as consistency and clarity in life roles and personal standards [112]. If anxiety is experienced on a long-term basis, this can result in increased social isolation that challenges one's self-identity [91, 113]. People with Parkinson's have a higher risk of losing their ability to participate in meaningful life roles, which fosters vulnerability towards social isolation and can negatively influence their self-identity [30, 114]. This suggests that PWPs experiencing anxiety may be at increased risk of losing their self-identity; this was reflected in the participants' stories.

Interwoven with changing self-identity were the participants' experiences of changing relationships with others, and how this amplifies the loss of self-identity. This included significant others, wider family and other people:

"...as I say I've always been a sociable outgoing person but, um, I think twice now before I accept an invitation. Or going out to dinner with friends I feel more, more uncertain and insecure than I ever have done and I usually still go, but I'm looking for an excuse quite often."

*(Clare, lines 149–155)*

"And those who, well my wife is increasingly in role of carer with a capital c rather than a small c."

*(Edward, lines 320–323)*

The participants' experiences suggest that these changing relationships are in part due to the change in physical functioning brought about by Parkinson's, as well as anxiety. Perhaps, as the condition increasingly affected the participant's ability to participate in the world, making them more static, it made it more effortful and challenging for them to adjust and respond to unexpected demands [29]. This may alter their relationships with others, which further challenges self-identity [112, 114]:

*"You can't help it and then it leaves you feeling as a man, are these things going to be haunting you forever more. Then you can have performance anxiety because of the Parkinson's and not sleep right. Then you've got the business of how does your wife feel about that. You've got the problem of being able to walk with her and hold her hand or be able to lay your hand across her body or something without the hand shaking. And, it-it's really unpleasant and that causes anxiety, definitely. Definitely. You just wish you could, you just wish you could go to sleep and wake up and you're ok."*

*(Alan, lines 614–628)*

Problems related to intimacy and sexual dysfunctions are a frequently recorded complication for PWPs but remain poorly investigated [115]. It can lead to relationship dysfunction and subsequent breakdown [116]. Alan expresses how this challenges his identity as a man, as well as a husband. Gender identity issues are also under-explored, with some suggesting that this may be a manifestation of hypersexuality i.e. sexual dysfunction leading to excessive sexual demand [117]. Thinking reflexively, it was interesting that Alan raised intimate relationships and illustrated his feelings about this in depth, yet this was not a topic identified by the researcher or in the literature review. As the researcher did not question other participants around this, the findings are limited regarding this and further research is warranted.

Previous research has suggested that anxiety primarily affects motor performance as a result of medication 'wearing-off' [118]. The stories shared by participants suggest a more complex relationship between anxiety and motor symptomology that cannot be explained by 'wearing-off' phenomenon alone [9]. The link between anxiety and physical symptoms in Parkinson's has previously been explored in the context of freezing of gait [21, 118]. There is less evidence regarding how anxiety affects other physical symptoms, such as pain and tremor in Parkinson's [9, 94] yet these are a vivid component of the participants' experiences. Freezing of gait (and the falls related with it) is a primary concern of healthcare professionals and researchers [31]. People with Parkinson's highlight their other symptoms as often being equal to, if not more important to, falls [32]. Therefore, a shift in perspective may be required so that healthcare professionals and researchers are more aware of the priorities of PWPs.

The findings of the theme "Amplification" align with other studies exploring the experience of Parkinson's. For example, Sunvisson [29] highlights how increased situational anxiety can result in a "closed down" (p.96) experience, possibly leading to reduced drive to socialise by PWPs [95]. However, Wressle et al. [30] suggest that the effects on social and public life may not solely be attributable to anxiety in Parkinson's. Adverse effects on social and public life may occur irrespective of anxiety in Parkinson's but may not have had such a high impact [119]. It could also be that the participants are experiencing slowed cognitive processing that is impacting on social interactions. Mild cognitive deficits can be detected in PWPs even before motor symptoms appear [120] this may be a contributing factor in this experience.

The participants identified anxiety as a negative experience that had a detrimental impact on their quality of life, echoing the findings of the patient and public involvement consultation [34]. Anxiety is overwhelmingly seen as a negative entity in Parkinson's literature, associated with issues such as fear of falling, and uncertainty in family and employment roles [8, 27, 94]. The participants' experiences reflect this, placing the negative experience as an aspect within the lived experience of anxiety in Parkinson's [36, 37].

Interestingly, this experience of anxiety was not the only experience of the participants. Beth strongly felt that her anxiety experience was not negative and indeed she gained positives from it:

*"Anxiety in a way helps because I don't do silly things. I mean I was doing silly things before. I was taking risks. And anxiety of keeping myself intact is important, it makes me be careful. I've got to think about the next move. I sometimes get in position and think, now they make me have this on <points to pendant alarm> that I don't want to call them whereas before I was falling stupidly all over the place and being silly about things, now my anxiety is useful. It's not pleasant. It's very unpleasant but it's useful."*

*(Beth, lines 406–418)*

During the member checking process, Beth voiced this feeling further, explaining that her anxiety gave her a stronger sense of self-identity as it drove her to cope. During the member checking process, Beth likened it to the *"climb of Everest on a daily basis"*. As the only participant reporting a lifetime history of anxiety, it is unclear if this reflects the lived experience of anxiety in Parkinson's or a unique experience to Beth; a further indication that a person-centred approach will be needed in an intervention. However, Bower et al. [121] report that pre-existing anxiety is likely a predictor of increased Parkinson's risk, meaning many more PWPs may experience anxiety before the condition develops. There is little literature exploring positive perceptions and implications of anxiety. Kashdan et al. [122] suggest that social anxiety may play a role in shaping positive experiences and events. In Parkinson's populations, Sotgiu and Rusconi [123] suggest that investigating the positive experiences of emotional events, such as anxiety, could lead to a greater understanding of the complex emotional landscape experienced by PWPs. Beth's story seems to have revealed a new facet of the lived experience of anxiety in Parkinson's that it would be beneficial to research further.

The findings of the theme "Anxiety is a funny thing" are echoed in other literature exploring experiences in Parkinson's [29, 30]. Fluctuating physical symptoms and their impact on independence and safety have previously been associated with causing anxiety in PWPs [27, 30]. Though Sunvisson [29] suggested that the sense of loss of control manifests in late-stage Parkinson's, the participants' stories reveal that this is also evident in their experience of the early stages of the condition.

All participants reported anxiety about living with a progressive condition with an unpredictable course. This fear of the future was an important aspect of the participants' perception and lived experience of anxiety:

*"And it's it's always a constant thing in the back of your mind what's going to happen in the future which also makes you think about things in a different light as well."*

*(Alan, lines 221–224)*

*"I can see depending on the way the condition goes I could be more and more anxious about me and those around me. But I hope to hang on to the fact that it's the way it is."*

*(Edward, lines 325–329)*

This has previously been identified in studies exploring the experience of Parkinson's, including anxiety [30]. It could be that the fear of future is linked to worries wider than just physical deterioration, such as being unable to support a relative if they become sick or uncertainty regarding family and work roles [30, 124].

People with Parkinson's who were involved in the patient and public involvement consultation that underpinned this study also expressed the ever-present nature of anxiety [34]. Due to the pathophysiology of Parkinson's, it could be that the participants experience this feeling of

omnipresent anxiety due to irreversible changes in brain biology [102, 125]. As previously cited, neurobiological theories for anxiety in Parkinson's remain unconfirmed while psychosocial factors have been clearly expressed [93, 114, 126]. As a tentative suggestion, perhaps the underlying brain pathology of the condition coupled with the psychosocial factors identified in the literature means that PWPs are more vulnerable to experiencing the feeling of anxiety as ever-present [29, 114, 125].

While a neurological basis for anxiety in Parkinson's is theorized, other factors have been articulated by PWPs [102, 114]. This suggests that interventional studies should move away from investigating strictly neurobiological solutions and encompass psychosocial and situational factors. Alan expressed this stance:

*"And you can always be given tablets for things but that that isn't the answer it's about being able to overcome. . ."*

*(Alan, lines 239–242)*

**Proposed model of the experience of PWPs experience of anxiety.**   This study's findings, alongside the existing literature, suggest that PWPs inhabit a lifeworld where anxiety is a multi-faceted experience that involves finding ways to cope, experiencing amplified symptoms, and is experienced in myriad ways. The preexisting literature of Parkinson's-associated dopamine deficiency and the findings of this study suggest these facets are shaped by the individual's neurobiology, experiences, and life context. As discussed in the introduction, the role of dopamine-deficiency in Parkinson's contributes to a neurobiological underpinning to anxiety in Parkinson's. Additionally, lifespan views of anxiety disorders espouse the importance of how individual experiences and context throughout life shape the manifestation of anxiety [127]. Considering all of these components, a lifeworld-led approach would be justified to ground care in human terms to promote participant health and well-being [128, 129].

In line with a lifeworld-led approach to care, a model of living with Parkinson's is represented in Fig 1. Future interventions need to focus on the component parts to help a person live well with Parkinson's. The model should not be understood in a deterministic way [52]. The components of the model may be differently proportioned depending on a person's experience of living with Parkinson's, which suggests assessment and intervention will need to be person centered. In terms of the individual components, for example, finding ways to cope suggests that self-management strategies may be a valued intervention within this population because of the importance PWPs place on being self-reliant in managing and coping with their anxiety. The next step is to explore whether this model holds true for the wider population of PWPs as a precursor to developing an intervention.

**Limitations of the study.**   This research has been conducted in line with American Psychological Associations' Journal Article Reporting Standards for Qualitative Research [46]. Its methodological strengths include a clear audit trail and the use of member checking and peer review. Despite this, it has limitations. Whilst the maximum variation criteria were met in terms of the experience of Parkinson's, the sample was limited to a particular age range, social class, sexual orientation, and ethnic group. That is, the sample consisted of white, heterosexual, English speaking people in the South West of England who were financially stable and lived in their own home. As Parkinson's affects people across all backgrounds, the study's findings should be seen as an overture, and further work is required to apprehend the complete picture of the lived experience of anxiety in Parkinson's.

## Conclusion

This study is, as far as we are aware, the first of its kind to explore the lived experience of anxiety in Parkinson's. The findings, although preliminary, have illuminated the complexity of the lived experience of anxiety for PWPs. Finding ways to cope is an important aspect of the anxiety experience, with value placed on self-reliance. It is important to recognise that not all participants found sharing their experience with others to be either positive or beneficial. People with Parkinson's may experience anxiety in Parkinson's as an amplification of symptoms; not only their physical signs but also their cognitive and emotional symptoms. This can often be more important to them; an important consideration particularly in hospitals where the management of physical symptoms usually takes precedence. It is essential that self-identity is considered with a deeper meaning, expanding beyond those of family and social roles. Anxiety is experienced in myriad ways both negatively and positively by PWPs. It is viewed as an ever-present, negative influence but can also be viewed as a positive force for PWPs to initiate action. This may require a shift in perspective of healthcare professionals changing their beliefs about anxiety in Parkinson's and, in turn, how they provide person-centred care to this group.

Due to the limitations identified, the findings of this study should be considered preliminary and an invitation for further research to explore the lived experience of anxiety in Parkinson's in richer depth. The multiple facets of the lived experience provide a basis for further research and this intelligence could then be used to develop an intervention for PWPs that focuses on their experiences rather than healthcare professional or researcher assumptions. Indeed, this study could be considered an open invitation to researchers working with those people with other long-term health conditions to explore the lived experience of anxiety. This could lead more collaborative working with those living with long-term health conditions to explore solutions that enable them to live their lives in the way of their choosing.

## Supporting information

**S1 File.**
(DOCX)

**S2 File.**
(DOCX)

**S3 File.**
(DOCX)

**S4 File.**
(DOCX)

**S5 File.**
(DOCX)

**S6 File.**
(DOCX)

**S7 File.**
(DOCX)

**S8 File.**
(DS_STORE)

**S1 Data.**
(NVPX)

## Acknowledgments

The authors would like to thank the participants for sharing their experiences and their time, all of the patient and public involvement consultees to contributing to the design of this study, Mr Ben Elliott for his help in the consent process, and Parkinson's UK for their support with both the patient and public involvement consultation and study recruitment.

## Author Contributions

**Conceptualization:** Christopher J. Lovegrove, Katrina Bannigan.

**Data curation:** Christopher J. Lovegrove.

**Formal analysis:** Christopher J. Lovegrove.

**Investigation:** Christopher J. Lovegrove.

**Methodology:** Christopher J. Lovegrove, Katrina Bannigan.

**Project administration:** Christopher J. Lovegrove.

**Software:** Christopher J. Lovegrove.

**Supervision:** Katrina Bannigan.

**Validation:** Katrina Bannigan.

**Writing – original draft:** Christopher J. Lovegrove.

**Writing – review & editing:** Christopher J. Lovegrove, Katrina Bannigan.

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
