## [Decision Letter · Decision Letter 0]

16 Apr 2020

PONE-D-20-03238

What is the lived experience of anxiety for people with Parkinson’s? A phenomenological study

PLOS ONE

Dear Mr Lovegrove,

Thank you for submitting your manuscript to PLOS ONE. After careful consideration, we feel that it has merit but does not fully meet PLOS ONE’s publication criteria as it currently stands. Therefore, we invite you to submit a revised version of the manuscript that addresses the points raised during the review process.

Please pay particular attention to revising ethnic categorisations.  Please provide more support for the use of a phenomenological approach.

We would appreciate receiving your revised manuscript by 17 May 2020. To enhance the reproducibility of your results, we recommend that if applicable you deposit your laboratory protocols in protocols.io, where a protocol can be assigned its own identifier (DOI) such that it can be cited independently in the future. For instructions see: http://journals.plos.org/plosone/s/submission-guidelines#loc-laboratory-protocols

We look forward to receiving your revised manuscript.

Kind regards,

Rosemary Frey

Academic Editor

PLOS ONE

2. Please include additional information regarding the interview used in the study and ensure that you have provided sufficient details that others could replicate the study. For instance, if you developed an interview guide as part of this study and it is not under a copyright license more restrictive than CC-BY, please include a copy, in both the original language and English, as Supporting Information.

4. Your ethics statement must appear in the Methods section of your manuscript. If your ethics statement is written in any section besides the Methods, please move it to the Methods section and delete it from any other section. Please also ensure that your ethics statement is included in your manuscript, as the ethics section of your online submission will not be published alongside your manuscript.

Reviewers' comments:

Reviewer's Responses to Questions

**Comments to the Author**

1. Is the manuscript technically sound, and do the data support the conclusions?

Reviewer #1: Partly

Reviewer #2: Partly

2. Has the statistical analysis been performed appropriately and rigorously? 

Reviewer #1: N/A

Reviewer #2: N/A

3. Have the authors made all data underlying the findings in their manuscript fully available?

Reviewer #1: Yes

Reviewer #2: Yes

4. Is the manuscript presented in an intelligible fashion and written in standard English?

Reviewer #1: No

Reviewer #2: Yes

5. Review Comments to the Author

Reviewer #1: This is a very interesting area of research and it is important that it is published.

Specific comments:

Reference for your opening statement? and most common disease.

Sense Line 87? It has also been identified that excessive anxiety in

87 PWPs results in a higher frequency of motor fluctuations,8, 9 not the severity.

Explain this assertion that anxiety is experienced differently : Walsh and Bennett 4 highlight that PWPs experience more anxiety than people with multiple sclerosis. This suggests that these individuals experience anxiety differently to other populations, and that an earlier age of anxiety onset in Parkinson’s could be an indicator of higher levels of anxiety.

Line 121-122 – Punctuation

Line 151 & line 156 – ethnic description by colour is not broadly internationally accepted as a descriptor - can you re-code or classify ethnicity?

Line 151 – please clarify sentence – currently I am wondering if participants who were married were also in relationships with different sex partners.

Not a complete sentence: As PWPs can experience cognitive deficits, 171 36 which can hinder their ability to give informed consent regarding individual decisions. 37

Line 175 – for those readers not familiar with the Metal Act or UK healthcare, was there any validated tools as part of the metal capacity assessment - or perhaps detail this more.

Word form errors e.g., differences perspectives

General comments: The conflation of the results and the discussion in the final section makes the display of evidence to support the thematic analysis more difficult to follow. Please separate these two sections out. Using more sub-headings also would help the reader. It is quite a laborious read currently as you pick off each of the layers in your theme table, present each one, discuss each one and continue on to the next. Results presentation could be more succinct, combining and summarising other supporting literature and changing sentence structures for variation - (Example lines 256-259 - Sunvisson 21 observed the importance of the support of others

257 when encountering increasingly stressful situations. Rapee, Peters 60 recognised

258 that family support or significant others is not only a protective factor in anxiety but

259 also a positive influence in reducing distress and accessing support.

(Reformulation example) Relational support has been found to be important in stressful situations (21), having a protective role and assisting to reduce distress (60).

Use the discussion to bring the strands together to support your Figure 1 summary and decide if your data does provide sufficient support. There is a disconnect between Table 1 and Figure 1 which a discussion section would take care of. The article would do well to also be shorter in length.

Use of participant voice in the article – good to have a coding system so that the reader can see if the quotes are coming from different participants within your dataset. Eg lines 581, 587

Lines 257 Rapee?

Reviewer #2: This manuscript describes a study which takes a phenomenological approach to understanding the lived experience of anxiety in Parkinson’s. The literature review is thorough and clearly provides a rationale for this work, as anxiety when living with Parkinson’s is not yet well understood. The topic is important as it could inform future interventions to improve the quality of life of people with Parkinson’s and their families.

I like the study, but I think there is more that could be done to improve its quality. My biggest concern is that it purports to be a phenomenological study, but doesn’t really position itself well in that way. By this I mean, I think there needs to be a stronger rationale for taking a phenomenological approach, over other qualitative approaches. I agree that a phenomenological approach is absolutely appropriate and what is needed, but I think the authors could go further to better represent phenomenology, especially as it may be a new or unfamiliar approach to many readers of this journal.

I think more could be done in this direction at the beginning to introduce the principals of phenomenology and why they are best suited to answer the research question. I quite like the mix of results and discussion as those references to theoretical concepts and previous literature are incorporated in a way that brings deeper explanations to the findings. However, while the (largely) psychological theories cited are useful for making sense of the findings, I think there is room for introducing some more phenomenological concepts here. Particularly, I think the elements of the lifeworld would help to explore more deeply some participants’ experiences, e.g. in terms of sociability or identity. Also, I think there is some work around lifeworld-led care (care informed by phenomenological work) that could be drawn upon in making recommendations for interventions and future research. I have taken the liberty of including some references related to this, which I think the authors might find useful.

More detail on each section of the paper is below.

Abstract

Themes don’t emerge – Braun & Clarke paper – this belittles the work you put in as a researcher making sense of the data.

[Braun, V., Clarke, V. & Weate, P. (2016). Using thematic analysis in sport and exercise research. In B. Smith & A. C. Sparkes (Eds.), Routledge handbook of qualitative research in sport and exercise (pp. 191-205). London: Routledge. – accessible for download when googled]

Theme titles tend to work better if there is an experiential element to them, especially if the work is designed to be phenomenological. More on this below.

In myriad ways – rather than in a myriad of ways.

Data Availability

• Is it possible to make all data available without restriction. It’s important in work like this that the data are treated carefully to ensure there are no unexpected breaches of confidentiality, which can be particularly difficult to manage with data of this kind. [See – BPS statement: https://twitter.com/healthpsycleeds/status/1248333598840819712 available on twitter; Wiley has a useful page too: https://authorservices.wiley.com/open-research/open-data/index.html]

When you say “data” can be added, what are you referring to? Is this anonymised transcripts or something else? It is very important to be clear about what kind of data are shared, that appropriate permissions are in place, and that data are anonymised as far as is possible. Note: even full copies of anonymised transcripts can be a risk because anonymity can never be fully guaranteed.

Introduction

Line 84 – this is of concern

Line 87 – not severity – delete the

Freezing of gait needs some explanation, even just that this is a common motor problem experienced by PWP in case there are readers without background knowledge of PD.

The link to the unexpected finding about not being able to provide care needs a little more explanation or introduction. It’s difficult to know why this is relevant here.

Line 117 – I don’t think you need the word highlighted

Materials and method

I’m not sure there is sufficient justification here for using phenomenology. What I mean by that is, why phenomenology over another qualitative approach? The rationale provided is a basic rationale for a qualitative, exploratory approach, but not specifically a phenomenological approach. I think a little more detail is required here to illustrate, perhaps to a reader new to phenomenology, why this approach is particularly useful here.

Line 142 – Parkinson’s UK’s ‘Reseach Support Network’ – insert apostrophe after UK OR write as Parkinson’s UK Research Support Network.

What is meant by a de facto relationship with different sex partners? This requires definition.

There is some repetition in this section which can be deleted.

Was the Bevan interview scheduled referenced or was it used? Was it adapted or used in its original form? It’d be helpful to include the schedule that was used in a figure or table, citing Bevan.

I would like to see more detail on what is involved in the bracketing process as outlined by Chan, Fung and Chien. Bracketing will be new to some readers, but even to those familiar with/expert in phenomenology, there are many ways of engaging in bracketing and so further detail here would increase transparency.

As above, there are many phenomenological methods of analysis and while some readers may be familiar with Moustakas’ approach, I think it’s worth including some more detail about how this works in practice. What does it add, specifically, to the process or perhaps the underlying objective of the thematic analysis?

A definition of trustworthiness would be helpful. Also, it would be helpful to describe what a theme is – perhaps with the help of Braun & Clarke’s work or that of Moustakas so the reader can then see how a discussion of a theme would be helpful in creating a set of themes that adequately & appropriately represent participants’ lived experience.

Some of the above is explained by way of describing the consultation with participants, but it’d help to elaborate on the process within supervision too.

Results and discussion

I’m not sure the theme titles denote phenomenological themes. I would expect to see something of the quality of the experience within the theme title, as a heuristic would – something which points toward the meaning of the experience as lived rather than use of a simple descriptor.

I think the ‘in vivo’ extracts you use in the text offer that experiential quality a phenomenological analysis should have. I would therefore include these quotations as part of the theme titles, both here and in the abstract.

Table: I don’t think you need to include the note about the sub-categories which didn’t make the cut. That will then make it much clearer that these sub-categories were clustered to create the themes presented.

Line 216 – UK spelling: behaviour

Line 217 – different perspectives – not differences

Please attribute quotations to a participant code/pseudonym. That we know if they’re male or female & can see how many contributed to each theme in different ways, thus building up a picture of each individual as we move through the results.

Line 264 – no need for possessive apostrophe – participants – just plural

I wonder if it’s also worth mentioning a sense of belonging or perhaps a sense of not being alone – as well as the upward/downward social comparisons. These are ‘valid’ interpretations, but don’t really speak to the phenomenological approach being taken. I think reference to some phenomenological constructs here (& elsewhere) would be helpful (e.g. Heidegger’s notion of being-with).

Line 439 – important – not importance

In discussing the sense of identity and changes in roles (from partner to carer, carer to Carer) I think there’s a missed opportunity to bring in the phenomenological elements of the lifeworld (here & elsewhere, but here, specifcially), e.g. sociability, sometimes referred to as intersubjectivity. As suggested above, I think the analysis works well & I like the reference to existing literature & theoretical explanations, but I think there’s a need to consult a more phenomenological literature in doing this for it to be more coherent with the methodological approach taken.

Line 521 – no apostrophe, participants is just plural, not possessive

I think in the final theme of perceptions it’s worth thinking about the role of meaning and meaninglessness in experiences of anxiety with PD. Meaninglessness can lead to a feeling of powerlessness; it might be that meaninglessness or uncertainty which comes with Parkinson’s which contributes to anxiety. Also, this would offer another opportunity to bring in a more phenomenological reading of the participants’ accounts.

Also, when discussion psychosocial interventions, there’s a missed opportunity here to talk of the need for a person-centred approach, or indeed a lifeworld-led approach (see Galvin & Todres’ work) when developing interventions to help reduce anxiety among PWP.

I think the findings related to the positive experience or positive meanings of anxiety in Parkinson’s is really interesting and points towards some social gerontology work on the co-existence of vulnerability and strengths, dependence and independence, suffering and wellbeing, especially in older age or in chronic illness (Grenier). I think this applies very well here and could drawn upon.

This counters your later discussion of the focus in much research on PD that is neurology, neurological treatment, and essentially, a cure. This doesn’t help those living with PD now and it doesn’t help them come to terms with their own experience of it.

I think your model demonstrates this need for a multifaceted approach to PD very well. I really like it! I wonder if you could refer to that a little earlier in the discussion of your findings so that we build a picture of it as we read – rather than leading to it as a crescendo at the end. (I think some signposting to the model is necessary throughout the discussion because of the way you’ve structured it as results and discussion together.) Also, this multifaceted notion of living well with Parkinson’s and anxiety speaks to the need to focus on the lifeworld – all the elements in a person’s experience, not simply their neurobiology. I think this case could be made more strongly by the introduction of some phenomenological theory and by emphasising the need for a phenomenological approach to fully understand the lived experience of PWP and thus to develop interventions that will boost their quality of life.

Some papers you may wish to read to help with integrating the phenomenological constructs throughout are below:

Baars, J., & Phillipson, C. (2013). Connecting meaning with social structure: theoretical foundations. In J. Baarsm, J. Dohmen, A. Grenier & C. Phillipson (Eds.), Ageing, meaning and social structure: connecting critical and humanistic gerontology (pp. 11_30). Bristol: Polity Press.

Galvin, K., & Todres, L. (2011). Kinds of well-being: A conceptual framework that provides direction for caring. International Journal of Qualitative Studies on Health & Wellbeing, 6, 10362. doi: http://dx.doi.org/10.3402/qhwv6i4.10362 Galvin, K., & Todres, L. (2013). Caring and well-being: A lifeworld approach. London: Routledge.

Aston, L., Shaw, R.L., & Knibb, R.C. (2019). Preliminary development of proxy-rated quality-of-life scales for children and adults with Niemann-Pick type C. Quality of Life Research, 28(11), 3083-3092.

Smith, L.J. & Shaw, R.L. (2016). Learning to live with Parkinson’s disease in the family unit: an interpretative phenomenological analysis of well-being. Medicine, Health Care & Philosophy. DOI: 10.1007/s11019-016-9716-3.

COREC is of course are the standards used in medical and health research to appraise the quality of qualitative research. They are useful, but don’t always fit every approach to qualitative research, and I’m not sure they’re that useful for assessing phenomenological work. You may instead wish to cite the American Psychological Associations’ Journal Article Reporting Standards for Qualitative Research: https://apastyle.apa.org/jars/qualitative

6. PLOS authors have the option to publish the peer review history of their article (what does this mean?). If published, this will include your full peer review and any attached files.

Reviewer #1: No

Reviewer #2: No

---

## [Author Response · Author response to Decision Letter 0]

23 Aug 2020

Dear Rosemary Frey,

Response to Reviewers: What is the lived experience of anxiety for people with Parkinson’s? A phenomenological study (PONE-D-20-03238)

Thank you for your response to our article, following the peer review process. We appreciate the opportunity to submit a revised manuscript and are grateful for the positive feedback which was very encouraging. We have provided a point by point response to the points the reviewers suggested need to be addressed. 

Comments to the Author: Reviewer #1

• Reference for your opening statement? and most common disease.

Reference (de Lau and Breteler, 2006) has been added to text.

• Sense Line 87? It has also been identified that excessive anxiety in 87 PWPs results in a higher frequency of motor fluctuations,8, 9 not the severity.

Sentence rewritten to improve the sense of the sentence

• Explain this assertion that anxiety is experienced differently: Walsh and Bennett 4 highlight that PWPs experience more anxiety than people with multiple sclerosis. This suggests that these individuals experience anxiety differently to other populations, and that an earlier age of anxiety onset in Parkinson’s could be an indicator of higher levels of anxiety.

Sentence rewritten to be more explicit about the differences and reduce confusion. Additional detail about why people with Parkinson’s experience anxiety differently has also been added.

• Line 121-122 – Punctuation

It was not clear to the authors what punctuation was being referred to so no change has been made. Please can you clarify what the specific issue is.

• Line 151 & line 156 – ethnic description by colour is not broadly internationally accepted as a descriptor - can you re-code or classify ethnicity?

We have assumed this is referring to the reference to the word white but we are not completely sure what you are driving at here. In https://www.ethnicity-facts-figures.service.gov.uk/style-guide/ethnic-groups white is used as a term in the list of ethnic groups. Have we misunderstood what you are driving at?

• Line 151 – please clarify sentence – currently I am wondering if participants who were married were also in relationships with different sex partners.

The sentences related to this have been amended to address this comment.

• Not a complete sentence: As PWPs can experience cognitive deficits, 171 36 which can hinder their ability to give informed consent regarding individual decisions. 37

This has been amended.

• Line 175 – for those readers not familiar with the Metal Act or UK healthcare, was there any validated tools as part of the metal capacity assessment - or perhaps detail this more.

A reference has been added to the text in response to this comment.

• Word form errors e.g., differences perspectives

The text has been amended to remove this error. Other errors, identified when re-reading of the text (and in response to reviewer 2 ‘s comments), have also been amended.

• General comments: The conflation of the results and the discussion in the final section makes the display of evidence to support the thematic analysis more difficult to follow. Please separate these two sections out. Using more sub-headings also would help the reader. It is quite a laborious read currently as you pick off each of the layers in your theme table, present each one, discuss each one and continue on to the next. Results presentation could be more succinct, combining and summarising other supporting literature and changing sentence structures for variation - (Example lines 256-259 - Sunvisson 21 observed the importance of the support of others 257 when encountering increasingly stressful situations. Rapee, Peters 60 recognised 258 that family support or significant others is not only a protective factor in anxiety but 259 also a positive influence in reducing distress and accessing support. (Reformulation example) Relational support has been found to be important in stressful situations (21), having a protective role and assisting to reduce distress (60).

It was hard to know how to respond to these comments because the conflation of findings and discussion is a convention that is widely used in reporting qualitative research. The second reviewer also liked this approach. We have maintained the approach of incorporating the findings and discussion but have tightened up the text so that there is greater clarity about the distinct parts.

The text has been changed to reflect the reformulation suggested.

• Use the discussion to bring the strands together to support your Figure 1 summary and decide if your data does provide sufficient support. There is a disconnect between Table 1 and Figure 1 which a discussion section would take care of. The article would do well to also be shorter in length.

In light of these comments, and reviewer 2’s comments, we have removed table 1 and formally identified figure 1 as a model with additional explanatory text. We hope this has resolved this issue.

• Use of participant voice in the article – good to have a coding system so that the reader can see if the quotes are coming from different participants within your dataset. Eg lines 581, 587

Lines 257 Rapee?

We have amended the text to reflect the names (pseudonyms) of the participants (and the transcript line numbers) associated with each quote to demonstrate that the responses are drawn from the full range of participants’ responses.

Lines 257 Rapee? – This is an author rather than a respondent. This should be clearer now that the participants names are cited after each quotation.

Comments to the Author: Reviewer #2

• I like the study, but I think there is more that could be done to improve its quality. My biggest concern is that it purports to be a phenomenological study, but doesn’t really position itself well in that way. By this I mean, I think there needs to be a stronger rationale for taking a phenomenological approach, over other qualitative approaches. I agree that a phenomenological approach is absolutely appropriate and what is needed, but I think the authors could go further to better represent phenomenology, especially as it may be a new or unfamiliar approach to many readers of this journal.

Thank you for your encouraging comments. We have rewritten the justification of the study to better represent it as a phenomenological study. We hope that we have gone into sufficient depth whilst still being concise. 

• I I quite like the mix of results and discussion as those references to theoretical concepts and previous literature are incorporated in a way that brings deeper explanations to the findings. However, while the (largely) psychological theories cited are useful for making sense of the findings, I think there is room for introducing some more phenomenological concepts here. Particularly, I think the elements of the lifeworld would help to explore more deeply some participants’ experiences, e.g. in terms of sociability or identity. Also, I think there is some work around lifeworld-led care (care informed by phenomenological work) that could be drawn upon in making recommendations for interventions and future research. I have taken the liberty of including some references related to this, which I think the authors might find useful.

This comment was very helpful in enabling us to revise the text. Our use of language detracted from the phenomenological approach in places, particularly the themes, and we have revised the text to more fully convey the experiential nature of the data.

• Abstract

Themes don’t emerge – Braun & Clarke paper – this belittles the work you put in as a researcher making sense of the data.

[Braun, V., Clarke, V. & Weate, P. (2016). Using thematic analysis in sport and exercise research. In B. Smith & A. C. Sparkes (Eds.), Routledge handbook of qualitative research in sport and exercise (pp. 191-205). London: Routledge. – accessible for download when googled]

Thank you for this observation. The text has been amended to convey a stronger sense of that a process of data analysis underpins this.

• Theme titles tend to work better if there is an experiential element to them, especially if the work is designed to be phenomenological. More on this below.

The themes have been revisited and we have revised how we have articulated them. The language has been amended in each theme to better convey the experiential element (including incorporating quotations from the data); we are grateful to both reviewers for pushing us to revisit this because this has enabled us to keep a strong focus on the lived experience.

• In myriad ways – rather than in a myriad of ways.

The text has been amended to reflect this change.

• Data Availability

Is it possible to make all data available without restriction? It’s important in work like this that the data are treated carefully to ensure there are no unexpected breaches of confidentiality, which can be particularly difficult to manage with data of this kind. [See – BPS statement: https://twitter.com/healthpsycleeds/status/1248333598840819712 available on twitter; Wiley has a useful page too: https://authorservices.wiley.com/open-research/open-data/index.html]

Having checked with both institutions, the data (anonymised with pseudonyms) can be made available without restriction alongside the NVivo data file.

• When you say “data” can be added, what are you referring to? Is this anonymised transcripts or something else? It is very important to be clear about what kind of data are shared, that appropriate permissions are in place, and that data are anonymised as far as is possible. Note: even full copies of anonymised transcripts can be a risk because anonymity can never be fully guaranteed.

Please see previous response. All transcripts have been anonymised and include no reference to the participants precise geographical location to help secure anonymity. The meta-data of the files have also been checked and anonymity is also preserved through this. This extends to the NVivo data file. Beyond the geographical location outlined in the article, there is no other reference that may risk participant anonymity.

• Introduction

Line 84 – this is of concern

Text amended.

Line 87 – not severity – delete the

Text amended.

Freezing of gait needs some explanation, even just that this is a common motor problem experienced by PWP in case there are readers without background knowledge of PD.

This is a good point and a lay definition has been added to the text.

• The link to the unexpected finding about not being able to provide care needs a little more explanation or introduction. It’s difficult to know why this is relevant here.

We agree when we read the text we could see that this looked a little incongruous and have amended the text. We hope the amendment makes more sense.

• Line 117 – I don’t think you need the word highlighted

Text amended.

• Materials and method

I’m not sure there is sufficient justification here for using phenomenology. What I mean by that is, why phenomenology over another qualitative approach? The rationale provided is a basic rationale for a qualitative, exploratory approach, but not specifically a phenomenological approach. I think a little more detail is required here to illustrate, perhaps to a reader new to phenomenology, why this approach is particularly useful here.

See comments above a fulsome justification has been added to the text.

• Line 142 – Parkinson’s UK’s ‘Research Support Network’ – insert apostrophe after UK OR write as Parkinson’s UK Research Support Network.

Text amended.

• What is meant by a de facto relationship with different sex partners? This requires definition.

See response to reviewer’s 1 comments where this has been addressed already.

• There is some repetition in this section which can be deleted.

We have edited the text to remove repetition.

• Was the Bevan interview scheduled referenced or was it used? Was it adapted or used in its original form? It’d be helpful to include the schedule that was used in a figure or table, citing Bevan.

Bevan has provided an approach to phenomenological interviewing; we did not use a schedule devised by him. We have made this clearer in the text and provided the interview schedule in supplemental materials.

• I would like to see more detail on what is involved in the bracketing process as outlined by Chan, Fung and Chien. Bracketing will be new to some readers, but even to those familiar with/expert in phenomenology, there are many ways of engaging in bracketing and so further detail here would increase transparency.

A definition of bracketing and a list of Chan, Fung and Chien’s strategies used in the study have been added to the text.

• As above, there are many phenomenological methods of analysis and while some readers may be familiar with Moustakas’ approach, I think it’s worth including some more detail about how this works in practice. What does it add, specifically, to the process or perhaps the underlying objective of the thematic analysis?

In revising the text we felt this had overly complicated what we did so have focussed solely on the thematic analysis that was conducted.

• A definition of trustworthiness would be helpful. 

This has been added.

• Also, it would be helpful to describe what a theme is – perhaps with the help of Braun & Clarke’s work or that of Moustakas so the reader can then see how a discussion of a theme would be helpful in creating a set of themes that adequately & appropriately represent participants’ lived experience.

A more detailed account of the data analysis has been added including a definition of a theme.

• Some of the above is explained by way of describing the consultation with participants, but it’d help to elaborate on the process within supervision too.

A more detailed account of the peer review process in supervision has been added.

• Results and discussion

I’m not sure the theme titles denote phenomenological themes. I would expect to see something of the quality of the experience within the theme title, as a heuristic would – something which points toward the meaning of the experience as lived rather than use of a simple descriptor.

See response to previous comment (above) where we addressed this issue.

• I think the ‘in vivo’ extracts you use in the text offer that experiential quality a phenomenological analysis should have. I would therefore include these quotations as part of the theme titles, both here and in the abstract.

See response to previous comment (above) where we addressed this issue.

• Table: I don’t think you need to include the note about the sub-categories which didn’t make the cut. That will then make it much clearer that these sub-categories were clustered to create the themes presented.

We agree that this is a better strategy and have removed the text.

• Line 216 – UK spelling: behaviour

Text amended.

• Line 217 – different perspectives – not differences

Text amended.

• Please attribute quotations to a participant code/pseudonym. That we know if they’re male or female & can see how many contributed to each theme in different ways, thus building up a picture of each individual as we move through the results.

This has been done using pseudonyms throughout; we have also added a note to the text explicitly stating that the names are pseudonyms.

• Line 264 – no need for possessive apostrophe – participants – just plural

Text amended.

• I wonder if it’s also worth mentioning a sense of belonging or perhaps a sense of not being alone – as well as the upward/downward social comparisons. These are ‘valid’ interpretations, but don’t really speak to the phenomenological approach being taken. I think reference to some phenomenological constructs here (& elsewhere) would be helpful (e.g. Heidegger’s notion of being-with).

We were not completely clear about what the expectations were here. The participants did not convey a sense of not being alone. Is it being suggested we should refer to literature related to this?

• Line 439 – important – not importance

Text amended.

• In discussing the sense of identity and changes in roles (from partner to carer, carer to Carer) I think there’s a missed opportunity to bring in the phenomenological elements of the lifeworld (here & elsewhere, but here, specifcially), e.g. sociability, sometimes referred to as intersubjectivity. As suggested above, I think the analysis works well & I like the reference to existing literature & theoretical explanations, but I think there’s a need to consult a more phenomenological literature in doing this for it to be more coherent with the methodological approach taken.

Again we are not completely sure what is being driven at here. We would welcome further clarity.

• Line 521 – no apostrophe, participants is just plural, not possessive

Text amended.

• I think in the final theme of perceptions it’s worth thinking about the role of meaning and meaninglessness in experiences of anxiety with PD. Meaninglessness can lead to a feeling of powerlessness; it might be that meaninglessness or uncertainty which comes with Parkinson’s which contributes to anxiety. Also, this would offer another opportunity to bring in a more phenomenological reading of the participants’ accounts.

Thank you for this insight- extra details have been added.

• Also, when discussion psychosocial interventions, there’s a missed opportunity here to talk of the need for a person-centred approach, or indeed a lifeworld-led approach (see Galvin & Todres’ work) when developing interventions to help reduce anxiety among PWP.

We have introduced this at various points in the paper and firmly linked it to the proposed model. This was a helpful addition that has helped us draw out something that was implicit in our previous draft.

• This counters your later discussion of the focus in much research on PD that is neurology, neurological treatment, and essentially, a cure. This doesn’t help those living with PD now and it doesn’t help them come to terms with their own experience of it.

We have amended this text to improve clarity and hope that it now makes more sense.

• I think your model demonstrates this need for a multifaceted approach to PD very well. I really like it! I wonder if you could refer to that a little earlier in the discussion of your findings so that we build a picture of it as we read – rather than leading to it as a crescendo at the end. (I think some signposting to the model is necessary throughout the discussion because of the way you’ve structured it as results and discussion together.) Also, this multifaceted notion of living well with Parkinson’s and anxiety speaks to the need to focus on the lifeworld – all the elements in a person’s experience, not simply their neurobiology. I think this case could be made more strongly by the introduction of some phenomenological theory and by emphasising the need for a phenomenological approach to fully understand the lived experience of PWP and thus to develop interventions that will boost their quality of life.

We have expanded on the proposed model so hopefully how it is presented now draws together the strands of the paper (and also meets with reviewer 1’s comments about the presentation of the findings).

• Some papers you may wish to read to help with integrating the phenomenological constructs throughout are below:

Baars, J., & Phillipson, C. (2013). Connecting meaning with social structure: theoretical foundations. In J. Baarsm, J. Dohmen, A. Grenier & C. Phillipson (Eds.), Ageing, meaning and social structure: connecting critical and humanistic gerontology (pp. 11_30). Bristol: Polity Press.

Galvin, K., & Todres, L. (2011). Kinds of well-being: A conceptual framework that provides direction for caring. International Journal of Qualitative Studies on Health & Wellbeing, 6, 10362. doi: http://dx.doi.org/10.3402/qhwv6i4.10362 Galvin, K., & Todres, L. (2013). Caring and well-being: A lifeworld approach. London: Routledge.

Aston, L., Shaw, R.L., & Knibb, R.C. (2019). Preliminary development of proxy-rated quality-of-life scales for children and adults with Niemann-Pick type C. Quality of Life Research, 28(11), 3083-3092.

Smith, L.J. & Shaw, R.L. (2016). Learning to live with Parkinson’s disease in the family unit: an interpretative phenomenological analysis of well-being. Medicine, Health Care & Philosophy. DOI: 10.1007/s11019-016-9716-3.

Thank you for being so supportive. 

• COREC is of course are the standards used in medical and health research to appraise the quality of qualitative research. They are useful, but don’t always fit every approach to qualitative research, and I’m not sure they’re that useful for assessing phenomenological work. You may instead wish to cite the American Psychological Associations’ Journal Article Reporting Standards for Qualitative Research: https://apastyle.apa.org/jars/qualitative

We have amended the text accordingly. 

Journal requirements: Additional requirements

The manuscript has been amended to meet the style requirements. The supplementary information file has been renamed accordingly. It is unclear how we need to name the main manuscript file as we have been unable to find this information on the PLOS One website. We have named in after the manuscript number. Please let us know if this is adequate or how the file needs to be named.

2. Please include additional information regarding the interview used in the study and ensure that you have provided sufficient details that others could replicate the study. For instance, if you developed an interview guide as part of this study and it is not under a copyright license more restrictive than CC-BY, please include a copy, in both the original language and English, as Supporting Information.

We have included the interview schedule as part of the supporting information

3. We note that you have stated that you will provide repository information for your data at acceptance. Should your manuscript be accepted for publication, we will hold it until you provide the relevant accession numbers or DOIs necessary to access your data. If you wish to make changes to your Data Availability statement, please describe these changes in your cover letter and we will update your Data Availability statement to reflect the information you provide. Data can now be made fully available. The cover letter has now been changed to reflect this.

4. Your ethics statement must appear in the Methods section of your manuscript. If your ethics statement is written in any section besides the Methods, please move it to the Methods section and delete it from any other section. Please also ensure that your ethics statement is included in your manuscript, as the ethics section of your online submission will not be published alongside your manuscript.

The ethics statement is in the methods section of our manuscript. We have embedded so that it is in the heart of this section rather than at the end.

---

## [Decision Letter · Decision Letter 1]

22 Sep 2020

PONE-D-20-03238R1

What is the lived experience of anxiety for people with Parkinson’s? A phenomenological study

PLOS ONE

Dear Dr. Lovegrove,

Thank you for submitting your manuscript to PLOS ONE. After careful consideration, we feel that it has merit but does not fully meet PLOS ONE’s publication criteria as it currently stands. Therefore, we invite you to submit a revised version of the manuscript that addresses the points raised during the review process.

Please correct the typos noted by Reviewer 2.

We look forward to receiving your revised manuscript.

Kind regards,

Rosemary Frey

Academic Editor

PLOS ONE

Reviewers' comments:

Reviewer's Responses to Questions

**Comments to the Author**

1. If the authors have adequately addressed your comments raised in a previous round of review and you feel that this manuscript is now acceptable for publication, you may indicate that here to bypass the “Comments to the Author” section, enter your conflict of interest statement in the “Confidential to Editor” section, and submit your "Accept" recommendation.

Reviewer #1: All comments have been addressed

Reviewer #2: All comments have been addressed

2. Is the manuscript technically sound, and do the data support the conclusions?

Reviewer #1: Yes

Reviewer #2: Yes

3. Has the statistical analysis been performed appropriately and rigorously? 

Reviewer #1: N/A

Reviewer #2: N/A

4. Have the authors made all data underlying the findings in their manuscript fully available?

Reviewer #1: Yes

Reviewer #2: Yes

5. Is the manuscript presented in an intelligible fashion and written in standard English?

Reviewer #1: Yes

Reviewer #2: Yes

6. Review Comments to the Author

Reviewer #1: Thank you for your feedback and the changes made to the paper in response. Upon re-reading it, I can see the strength of your approach, especially with the changes made in the discussion around the proposed model of experience of PWP & anxiety. It is an interesting topic.

On the lines 120-121 (which I can not identify because I can no longer access the earlier draft) - but I think it will be this run-on sentence, which still exists in the text. Needs a semi-colon or a coordinating conjunction or a full stop after "day".

Although PWPs can be treated with dopamine-replacement medication they often experience a

marked increase in symptoms as the medication wears off throughout the day [16],

primary], primary (or type-1) worry can in turn rapidly progress to type-2 worry (metaworry,

or ‘worry about worry’).

Reviewer #2: I know there were a few comments for which authors requested further clarity, however, I think the changes made are appropriate and in my view no further major changes are required.

I just noticed a few typos in the additional text

- page 30: end of paragraph 1: participant and [MISSING WORD] wellbeing

- page 30: second paragraph: understood in A [add] deterministic

- page 30: second paragraph: The components of THE [add] model

7. PLOS authors have the option to publish the peer review history of their article (what does this mean?). If published, this will include your full peer review and any attached files.

Reviewer #1: No

Reviewer #2: No

---

## [Author Response · Author response to Decision Letter 1]

26 Sep 2020

Dear Rosemary Frey,

Response to Reviewers: What is the lived experience of anxiety for people with Parkinson’s? A phenomenological study (PONE-D-20-03238)

Thank you for your response to our article, following the peer review process. We appreciate the opportunity to submit a revised manuscript. We have submitted the revised manuscripts as requested.

We would like to thank the reviewers for the time that they have taken to review our article. The feedback received has been positive in helping us to improve our article, for which we are grateful.

Yours sincerely,

The authors

---

## [Decision Letter · Decision Letter 2]

17 Nov 2020

PONE-D-20-03238R2

What is the lived experience of anxiety for people with Parkinson’s? A phenomenological study

PLOS ONE

Dear Dr. Lovegrove,

Thank you for submitting your manuscript to PLOS ONE. After careful consideration, we feel that it has merit but does not fully meet PLOS ONE’s publication criteria as it currently stands. Therefore, we invite you to submit a revised version of the manuscript that addresses the points raised during the review process.

Please address the methodological issues raised by Reviewer 3.

We look forward to receiving your revised manuscript.

Kind regards,

Rosemary Frey

Academic Editor

PLOS ONE

Reviewers' comments:

Reviewer's Responses to Questions

**Comments to the Author**

1. If the authors have adequately addressed your comments raised in a previous round of review and you feel that this manuscript is now acceptable for publication, you may indicate that here to bypass the “Comments to the Author” section, enter your conflict of interest statement in the “Confidential to Editor” section, and submit your "Accept" recommendation.

Reviewer #2: All comments have been addressed

Reviewer #3: (No Response)

2. Is the manuscript technically sound, and do the data support the conclusions?

Reviewer #2: Yes

Reviewer #3: Yes

3. Has the statistical analysis been performed appropriately and rigorously? 

Reviewer #2: N/A

Reviewer #3: N/A

4. Have the authors made all data underlying the findings in their manuscript fully available?

Reviewer #2: Yes

Reviewer #3: Yes

5. Is the manuscript presented in an intelligible fashion and written in standard English?

Reviewer #2: Yes

Reviewer #3: Yes

6. Review Comments to the Author

Reviewer #2: I think the authors have dealt well with the comments from both reviewers. I don’t think there is any need to make further changes on the issues they requested clarity on. The amendments have improved the paper and improved the transparency of the phenomenological method.

Reviewer #3: I believe this study is indeed an important contribution to the phenomenological understanding of Parkinson’s disease. I find its ability to connect what can sometimes be an abstract field of study to clinical intervention as an important example for future work. However, I do believe that the study would benefit from further refinement – particularly as it concerns some of the study’s philosophical and technical positions. Please see the comments below for my suggestions to the authors:

1) My main concern is with the study’s description of its methodology. I am unsure if the reader can understand it as a cohesive whole and if it is conveyed in a way that would allow future researchers to replicate the study’s methodology. Below are several points regarding this matter:

a. The study cites several sources from which it derives aspects of its methodology. However, the borrowed components are left vague. Assuming that you are using a somewhat novel approach (i.e., not “one” comprehensive model like the “Phenomenological Psychological Method [Giorgi, & Giorgi, 2003], the Interpretive Phenomenological Analysis [Smith, & Shinebourne, 2012], etc.), the reader requires further explanation regarding rational and execution of these different aspects. This can be done similarly to how the author explained how the thematic analysis was conducted. The following questions should be taken into consideration – for most a sentence or two should suffice:

i. How does one “bracket” according to Chan et al? (pg. 6) You name some of the steps in the process, but the reader is not sure how you implemented them or what they necessarily mean.

ii. How did you use Bevan to create the interview? Or what suggestions did you follow? (pg. 8)

iii. What is the “immersion phase” and where does it come from? (pg. 9)

iv. You mention using NVivo v10 – what does the software do? Some readers will be unfamiliar with this. (pg. 9)

v. Where does the term “Trustworthiness” come from? (pg. 9) Is this a technical term or did you coin it? This seems similar to the hermeneutic circle (See comments below re: hermeneutics).

vi. Lastly, perhaps it is worth the authors mentioning how the above techniques fit together under a descriptive phenomenological framework. Or perhaps brief elaborations on the above-mentioned points will make this self-evident.

2) Another concern is some ambiguity or confusion on study’s position as representing “descriptive phenomenology.” The study seems to claim to be descriptive (i.e. presenting a phenomenon without using foreknowledge or interpretation) while also offering some of the author’s interpretations and contextualization, which is more akin to a hermeneutic or interpretive stance. The following are several points that warrant attention:

a. If this study is to be considered a “descriptive phenomenology” the authors should seriously consider separating out the descriptions of the themes they found from any interpretations of such themes. Some instances include:

i. The use of psychological concepts of “upward and downward comparison” to explicate coping mechanisms. (pg. 12)

ii. Offering explanation of adverse social experiences via cognitive psychology (i.e. “slowed cognitive processing”). (pg. 17) This might be remedied by tying this theme to your findings that subject report “inability to think,” as opposed to the author’s cognitive explanation.

iii. Interpreting the theme of “Ability to think” through a cognitive explanation (i.e. overload of working memory). (pg. 18)

iv. Interpreting emotion experience through physiological framework (i.e., dopaminergic pathways). (pg. 19)

v. Introducing powerlessness/meaninglessness to extrapolate when discussing subjects’ experience of loss of control. (pg. 26) This does not seem to come directly from subject reports.

vi. Using pathophysiology to explain omnipresent anxiety. (pg. 29)

b. One possible solution to the issue above is for the authors to separate out the interpretations of subjects’ accounts from the accounts themselves – possibly by differentiating “Results” section (i.e. descriptions of subject reports) from a “Discussion” section (i.e. elaborations, interpretations, and comparison of subject accounts to other fields of study or established work). I see that another reviewer suggested a similar edit, but I hope these points illuminate some of the theoretical reasons why one might consider doing so. To clarify, I believe these interpretations and comparisons are useful and have good insight, but to be truly “descriptive” one should rely more purely on subject reports.

c. In addition, it should be pointed out that the term “lifeworld,” which the study uses, is a term borrowed from hermeneutic, or interpretive phenomenology. The term is an effort by Heidegger to emphasize the need to contextualize a person’s experience, which is opposed to the aspirations of Husserl’s bracketing. This is not to say that the authors should do away with this word, as it has been widely adopted throughout phenomenological research, but rather it brings attention to some ambiguity in the study’s philosophical underpinnings.

d. For further overview on the difference between descriptive and interpretive phenomenology, I suggest looking at Lopez & Willis’ “Descriptive versus interpretive phenomenology: Their contributions to nursing knowledge” (2004).

3) The study overall would also benefit from further copy editing and general tightening up some of the prose throughout the paper. Two examples below:

a. Is the following a statement or a question? --- “Therefore, perhaps a shift in perspective is required so that healthcare professionals and researchers are more aware of the priorities of PWPs?” (pg. 23)

b. This sentence could benefit from a comma after “insights” and perhaps hyphenating “anxiety-specific” --- “Whilst these studies provide insights generally there is a lack of anxiety specific research in Parkinson’s.” (pg. 5)

Overall, I think this is a valuable piece of work. I hope you will consider the above revisions. There is indeed no one way to conduct phenomenology, but I believe taking these points into consideration will improve the study’s philosophical and methodological groundings.

7. PLOS authors have the option to publish the peer review history of their article (what does this mean?). If published, this will include your full peer review and any attached files.

Reviewer #2: No

Reviewer #3: No

---

## [Author Response · Author response to Decision Letter 2]

9 Feb 2021

Dear Rosemary Frey,

PONE-D-20-03238R2

What is the lived experience of anxiety for people with Parkinson’s? A phenomenological study

PLOS ONE

Please find attached our revised manuscript as a separate file labelled 'Revised Manuscript with Track Changes'. We are grateful to the reviewers for their comments, particularly their positive encouragement to revise the draft, and have summarised our responses in the table below.

Reviewers' comments Responses to Reviewers

Reviewer #2: I think the authors have dealt well with the comments from both reviewers. I don’t think there is any need to make further changes on the issues they requested clarity on. The amendments have improved the paper and improved the transparency of the phenomenological method.

 Thank you for the feedback. It is gratifying to know we have addressed your comments.

Reviewer #3: I believe this study is indeed an important contribution to the phenomenological understanding of Parkinson’s disease. I find its ability to connect what can sometimes be an abstract field of study to clinical intervention as an important example for future work.

 Thank you for this feedback which was helpful in preparing the revised manuscript.

Reviewer #3: I do believe that the study would benefit from further refinement – particularly as it concerns some of the study’s philosophical and technical positions.

My main concern is with the study’s description of its methodology. I am unsure if the reader can understand it as a cohesive whole and if it is conveyed in a way that would allow future researchers to replicate the study’s methodology. We have tried to address each of the comments identified and hope that in doing so we have made the methodology more cohesive and promoted the replicability of the study.

Reviewer #3: a. The study cites several sources from which it derives aspects of its methodology. However, the borrowed components are left vague. Assuming that you are using a somewhat novel approach (i.e., not “one” comprehensive model like the “Phenomenological Psychological Method [Giorgi, & Giorgi, 2003], the Interpretive Phenomenological Analysis [Smith, & Shinebourne, 2012], etc.), the reader requires further explanation regarding rational and execution of these different aspects. This can be done similarly to how the author explained how the thematic analysis was conducted.

 You are right we did not use one definitive approach to phenomenology and drew on a number of sources to inform our methodology. We hope the amendments we have made to different aspects highlighted means the text is more cohesive. We appreciate your openness to how we have approached phenomenology.

Reviewer #3: The following questions should be taken into consideration – for most a sentence or two should suffice:

i. How does one “bracket” according to Chan et al? (pg. 6) You name some of the steps in the process, but the reader is not sure how you implemented them or what they necessarily mean.

 We have annotated the text with explanations of Chan et al’s approach and explicitly stated that these steps were used in the study.

Reviewer #3: ii. How did you use Bevan to create the interview? Or what suggestions did you follow? (pg. 8)

 We have amended the text to explain that we adapted the wording with an example to bring this alive for the reader. 

Reviewer #3: iii. What is the “immersion phase” and where does it come from? (pg. 9)

 We have added in a succinct definition with a reference to the literature so it evident to the reader where the concept comes from.

Reviewer #3: iv. You mention using NVivo v10 – what does the software do? Some readers will be unfamiliar with this. (pg. 9)

 We have added in a definition for anyone unfamiliar with this software. 

Reviewer #3: v. Where does the term “Trustworthiness” come from? (pg. 9) Is this a technical term or did you coin it? This seems similar to the hermeneutic circle (See comments below re: hermeneutics).

 We have defined this and added in a reference.

Reviewer #3: vi. Lastly, perhaps it is worth the authors mentioning how the above techniques fit together under a descriptive phenomenological framework. Or perhaps brief elaborations on the above-mentioned points will make this self-evident.

 We hope the way we have elaborated on the text has made this self-evident. 

Reviewer #3: 2) Another concern is some ambiguity or confusion on study’s position as representing “descriptive phenomenology.” The study seems to claim to be descriptive (i.e. presenting a phenomenon without using foreknowledge or interpretation) while also offering some of the author’s interpretations and contextualization, which is more akin to a hermeneutic or interpretive stance.

 As with point one, we hope the amendments we have made to different aspects highlighted mean it is much clearer that this is a descriptive phenomenology. 

Reviewer #3: The following are several points that warrant attention:

a. If this study is to be considered a “descriptive phenomenology” the authors should seriously consider separating out the descriptions of the themes they found from any interpretations of such themes.

 This has been done.

Reviewer #3: Some instances include:

i. The use of psychological concepts of “upward and downward comparison” to explicate coping mechanisms. (pg. 12)

 As well as separating the findings and discussion, the text is amended to show what was described by the participants and how it relates to the existing literature. 

Reviewer #3: ii. Offering explanation of adverse social experiences via cognitive psychology (i.e. “slowed cognitive processing”). (pg. 17) This might be remedied by tying this theme to your findings that subject report “inability to think,” as opposed to the author’s cognitive explanation.

Reviewer #3: iii. Interpreting the theme of “Ability to think” through a cognitive explanation (i.e. overload of working memory). (pg. 18)

 The text has been reordered to resolve this confusion (see p26 in revised manuscript).

Reviewer #3: iv. Interpreting emotion experience through physiological framework (i.e., dopaminergic pathways). (pg. 19)

 The text has been reordered to resolve this confusion (see p26 in revised manuscript).

Reviewer #3: v. Introducing powerlessness/meaninglessness to extrapolate when discussing subjects’ experience of loss of control. (pg. 26) This does not seem to come directly from subject reports.

 We have moved this point to the discussion section and reworded to add clarity.

Reviewer #3: vi. Using pathophysiology to explain omnipresent anxiety. (pg. 29)

 We have linked this back to our introductory literature review in a way that we hope adds clarity.

Reviewer #3: b. One possible solution to the issue above is for the authors to separate out the interpretations of subjects’ accounts from the accounts themselves – possibly by differentiating “Results” section (i.e. descriptions of subject reports) from a “Discussion” section (i.e. elaborations, interpretations, and comparison of subject accounts to other fields of study or established work). I see that another reviewer suggested a similar edit, but I hope these points illuminate some of the theoretical reasons why one might consider doing so. To clarify, I believe these interpretations and comparisons are useful and have good insight, but to be truly “descriptive” one should rely more purely on subject reports.

 We have done this. We did not fully understand the import of what the previous reviewer was suggesting. Thanks you for the additional clarification here. 

Reviewer #3: c. In addition, it should be pointed out that the term “lifeworld,” which the study uses, is a term borrowed from hermeneutic, or interpretive phenomenology. The term is an effort by Heidegger to emphasize the need to contextualize a person’s experience, which is opposed to the aspirations of Husserl’s bracketing. This is not to say that the authors should do away with this word, as it has been widely adopted throughout phenomenological research, but rather it brings attention to some ambiguity in the study’s philosophical underpinnings.

 We appreciate your observation and acknowledge the ambiguity that could arise but have kept the concept in. This is because it was a recommendation in a previous review and we believe it does enhance the understanding of the lived experiences of the participants in this study.

Reviewer #3: d. For further overview on the difference between descriptive and interpretive phenomenology, I suggest looking at Lopez & Willis’ “Descriptive versus interpretive phenomenology: Their contributions to nursing knowledge” (2004).

 Thank you for this suggestion and we have incorporated this reference into the text.

Reviewer #3: 3) The study overall would also benefit from further copy editing and general tightening up some of the prose throughout the paper. Two examples below:

 We have tried to copy edit and tighten up the prose.

Reviewer #3: a. Is the following a statement or a question? --- “Therefore, perhaps a shift in perspective is required so that healthcare professionals and researchers are more aware of the priorities of PWPs?” (pg. 23)

 We have edited the text so it now reads – 

“Therefore, a shift in perspective may be required so that healthcare professionals and researchers are more aware of the priorities of PWPs.”

Reviewer #3: b. This sentence could benefit from a comma after “insights” and perhaps hyphenating “anxiety-specific” --- “Whilst these studies provide insights generally there is a lack of anxiety specific research in Parkinson’s.” (pg. 5)

 We agree that this improves the sense of the sentence and have made the edits suggested. 

Reviewer #3: Overall, I think this is a valuable piece of work. I hope you will consider the above revisions. There is indeed no one way to conduct phenomenology, but I believe taking these points into consideration will improve the study’s philosophical and methodological groundings.

 Thank you for being so encouraging. We hope the resultant edits have improved the study’s philosophical and methodological groundings.

We hope we have addressed the comments sufficiently in the revised manuscript to merit publication and we look forward to hearing from you in due course.

Kind regards, 

Chris Lovegrove

Katrina Bannigan

---

## [Decision Letter · Decision Letter 3]

18 Mar 2021

What is the lived experience of anxiety for people with Parkinson’s? A phenomenological study

PONE-D-20-03238R3

Dear Mr. Lovegrove,

We’re pleased to inform you that your manuscript has been judged scientifically suitable for publication and will be formally accepted for publication once it meets all outstanding technical requirements.

Kind regards,

Rosemary Frey

Academic Editor

PLOS ONE

Additional Editor Comments (optional):

Reviewers' comments:

Reviewer's Responses to Questions

**Comments to the Author**

1. If the authors have adequately addressed your comments raised in a previous round of review and you feel that this manuscript is now acceptable for publication, you may indicate that here to bypass the “Comments to the Author” section, enter your conflict of interest statement in the “Confidential to Editor” section, and submit your "Accept" recommendation.

Reviewer #2: All comments have been addressed

Reviewer #3: All comments have been addressed

2. Is the manuscript technically sound, and do the data support the conclusions?

Reviewer #2: Yes

Reviewer #3: Yes

3. Has the statistical analysis been performed appropriately and rigorously? 

Reviewer #2: N/A

Reviewer #3: Yes

4. Have the authors made all data underlying the findings in their manuscript fully available?

Reviewer #2: No

Reviewer #3: Yes

5. Is the manuscript presented in an intelligible fashion and written in standard English?

Reviewer #2: Yes

Reviewer #3: Yes

6. Review Comments to the Author

Reviewer #2: I am very happy with the revisions made. It is clear that you have responded to all the suggestions made. I am impressed with the paper and will look forward to seeing it published.

Reviewer #3: I believe the authors have sufficiently amended the text and incorporated reviewer recommendations. I do not think any further changes are needed. The revision submitted results in a paper with improved philosophical and methodological groundings. I believe the manuscript at hand is an important contribution to the phenomenological understanding of Parkinson's Disease and serves a sound example of employing phenomenological thought to arrive at clinical interventions.

7. PLOS authors have the option to publish the peer review history of their article (what does this mean?). If published, this will include your full peer review and any attached files.

Reviewer #2: No

Reviewer #3: No

---

## [Editor Report · Acceptance letter]

30 Mar 2021

PONE-D-20-03238R3 

What is the lived experience of anxiety for people with Parkinson’s? A phenomenological study 

Dear Dr. Lovegrove:

I'm pleased to inform you that your manuscript has been deemed suitable for publication in PLOS ONE. Congratulations! Your manuscript is now with our production department. 

Kind regards, 

on behalf of

Dr. Rosemary Frey 

Academic Editor

PLOS ONE